# Characterization of the LysP2110-HolP2110 Lysis System in *Ralstonia solanacearum* Phage P2110

**DOI:** 10.3390/ijms241210375

**Published:** 2023-06-20

**Authors:** Kaihong Chen, Yanhui Guan, Ronghua Hu, Xiaodong Cui, Qiongguang Liu

**Affiliations:** College of Plant Protection, South China Agricultural University, Guangzhou 510642, China; chenkaihong@stu.scau.edu.cn (K.C.); guan9917@126.com (Y.G.); dongbuguazhi@163.com (X.C.)

**Keywords:** plant bacterial wilt disease, *Ralstonia solanacearum*, lysins, biocontrol, holin, broad-spectrum antibacterial activity

## Abstract

*Ralstonia solanacearum*, a pathogen causing widespread bacterial wilt disease in numerous crops, currently lacks an optimal control agent. Given the limitations of traditional chemical control methods, including the risk of engendering drug-resistant strains and environmental harm, there is a dire need for sustainable alternatives. One alternative is lysin proteins that selectively lyse bacteria without contributing to resistance development. This work explored the biocontrol potential of the LysP2110-HolP2110 system of *Ralstonia solanacearum* phage P2110. Bioinformatics analyses pinpointed this system as the primary phage-mediated host cell lysis mechanism. Our data suggest that LysP2110, a member of the *Muraidase* superfamily, requires HolP2110 for efficient bacterial lysis, presumably via translocation across the bacterial membrane. LysP2110 also exhibits broad-spectrum antibacterial activity in the presence of the outer membrane permeabilizer EDTA. Additionally, we identified HolP2110 as a distinct holin structure unique to the *Ralstonia* phages, underscoring its crucial role in controlling bacterial lysis through its effect on bacterial ATP levels. These findings provide valuable insights into the function of the LysP2110-HolP2110 lysis system and establish LysP2110 as a promising antimicrobial agent for biocontrol applications. This study underpins the potential of these findings in developing effective and environment-friendly biocontrol strategies against bacterial wilt and other crop diseases.

## 1. Introduction

*Ralstonia solanacearum*, also known as a bacterial wilt pathogen, is a Gram-negative bacterium that causes devastating plant diseases worldwide [1]. This soil-borne pathogen leads to crops’ rapid wilting and death, ultimately resulting in crop failure. Currently, there is an urgent need to develop effective therapeutic agents for bacterial wilt disease, as traditional chemical agents are associated with bacterial resistance and harm to human health and the environment [2,3]. Bacteriophages are gaining increasing attention in the biological control of bacterial diseases due to their ability to specifically disrupt bacteria without causing drug resistance [4,5,6]. However, the use of bacteriophages is limited by several factors, including low biocompatibility, narrow applicability, pharmacokinetics, and low antibacterial activity [7].

Endolysin is a peptidoglycan hydrolase expressed by bacteriophages during the late stage of bacterial infection, playing a critical role in degrading the host bacteria’s peptidoglycan and releasing bacteriophage progeny [8]. This new type of antibacterial agent offers several evolutionary advantages over traditional chemical agents, including high efficiency in lysing activity, no resistance, and high biocompatibility [9].

In the typical two-component lysis system of bacteriophages, endolysin is synthesized by late gene expression during the lysis cycle [10] and accumulates harmlessly in the cytoplasm. However, it cannot penetrate the cell membrane and bind to the peptidoglycan layer without the assistance of a holin-hydrophobic integral membrane protein [11]. Holin forms raft-like aggregates on the bacterial inner membrane, causing nonspecific pores on the cell membrane [12], which allows endolysin to pass through and cut the peptidoglycan. As a result, the difference in osmotic pressure between the inside and outside of the cell leads to lysis and the release of bacteriophage progeny [13,14]. Therefore, holin plays a crucial role in controlling the exact time of cell lysis, ensuring the maximum efficiency of endolysin in lysing activity [15].

Some bacteriophages also contain anti-holin proteins, expressed by dual promoter sequences of holin genes. These proteins can form heterologous dimers with holin proteins to prevent the premature lysis of bacteria before bacteriophage assembly is completed [16]. However, there currently needs to be a clear conclusion on how bacteriophages that do not have anti-holin proteins regulate the precise timing of bacterial lysis. In addition, some bacteriophages have SAR-endolysin, which has a signal-blocking release (SAR) domain and can cross the membrane by the host bacterium’s Sec secretion system without relying on holin assistance, such as the endolysin phiKMV in bacteriophages P1 and P21 [17].

Previous research has shown that endolysin has significant potential for treating bacterial infections [18,19]. Researchers have established a large number of animal models, such as skin infection models [20], pneumonia infection models [21], vaginal infection models [22], and others. In these animal models, endolysin can quickly kill target bacteria, causing their numbers to decrease significantly in a short period of time. However, due to the differences in membrane structure between Gram-positive and Gram-negative bacteria, there needs to be more research on the application of endolysins in the bacterial treatment of plant diseases. In particular, there needs to be more research on the *Ralstonia* phage P2110 lysis system in relation to the treatment of bacterial wilt. This study aims to characterize the lysis system of *Ralstonia* phage P2110 and investigate the mechanism by which holins regulate the timing of bacterial lysis, providing insights into the use of endolysin-based biological control agents for the prevention and treatment of bacterial wilt in crops.

## 2. Results

### 2.1. Genome Analysis of Bacteriophage P2110

Bacteriophage P2110 revealed a total length of 59381bp, with 76 predicted ORFs. The ORFs can be divided into five main categories: hypothetical proteins, DNA replication and modification proteins, packaging proteins, lysis proteins, and structural proteins (Figure 1).

Phylogenetic analysis of the phage’s terminase large subunit (TerL) (Table 1) revealed that phage P2110 showed a high level of homology with *Ralstonia* phage GP4 (GenBank accession Number: NC_054964.1) and belonged to *Duplodnaviria, Heunggongvirae*, *Uroviricota*, *Caudoviricetes*, and *Gervaisevirus*, which is capable of infecting *Ralstonia solanacearum* (Figure 2). A genome comparison by BLAST analysis revealed that the genomic DNA sequence identity between P2110 and GP4 was 96.63%, with a query coverage of 86%.

### 2.2. Bioinformatics Analysis of the Phage P2110 Lysis System

Based on the bioinformatics analysis, we predicted that the lysis system of phage P2110 comprises two proteins: endolysin (LysP2110) and holin (HolP2110). The 23–196 amino acid region of LysP2110 contains the conserved domain of the Muraidase superfamily (Figure 3A), which has N-acetylglucosaminidase activity and cleaves the β-(1,4) glycosidic bond between the N-acetylglucosamine and N-acetylmuramic acid residues in the peptidoglycan skeleton. The 101st glutamic acid residue is a conserved catalytic residue. LysP2110 has a molecular weight of 22.2 kDa and lacks a signal peptide or transmembrane domain, indicating that it requires holin to cross the cytoplasmic membrane [23]. AlphaFold2 predicts the 3D structural model of LysP2110 (Figure 3B).

Initially, we could not predict the holin protein of phage P2110 using the NCBI and SMART online software. However, previous research has shown that the holin gene is usually located near the endolysin gene and forms a lysing cassette, with the protein containing at least one transmembrane domain [11]. Using ORF Finder and TMHMM to predict the gene near the endolysin, we identified a 76 amino acid protein with a transmembrane domain in the 13–35 amino acid region. Unlike typical lysing cassettes, the holin of phage P2110 is located downstream of the endolysin gene and lacks a double start sequence [24]. Interestingly, the SignalP analysis revealed that the N-terminus of the holin protein contains an SP (Sec/SPI)-type signal peptide structure, which contrasts with the holin structure reported in previous studies.

Finally, we determined the structure of HolP2110 by the bioinformatics analysis of homologous proteins (Figure 4A), amino acid hydrophobicity analysis, and protein 3D model prediction (Figure 4B). The 1–30 amino acid region of HolP2110 contains a signal peptide structure, and the 39–61 amino acid region of HolP2110 contains a transmembrane domain (Figure 3A). Presumably, this is a structure unique to HolP2110.

### 2.3. HolP2110 Is a Specific Holin Structure in the Ralstonia Phage

Based on the analysis of the phylogenetic tree of the bacteriophage P2110, we conducted a comparative analysis of the genomes of bacteriophages closely related to P2110. It was found that these bacteriophages all contain a highly similar protein to HolP2110 and that they share the same structure. A signal peptide is present at amino acids 1–30, while amino acids 39–61 have a transmembrane structural domain (Figure 4A). Using AlphaFold2 to predict the 3D structure of the protein, we found that HolP2110 mainly comprises two large helix structures, which likely play a role in localizing holin to the inner membrane in relation to the formation of late nonspecific pores (Figure 4B). In some bacteriophages, these proteins are still annotated as hypothetical proteins. Therefore, we have named this class of proteins HolP2110-like holins.

Further analysis of the conserved functional domains of HolP2110 using NCBI (Conserved Domains) revealed that its structure does not belong to any of the 65 holin families previously reported [25]. Additionally, when we performed a Blastp analysis on the HolP2110 protein sequence, the results showed that this type of holin protein is only annotated as a holin in a few *Ralstonia* phages, while in others, it is labeled as a putative bacterial protein. Finally, we conducted a phylogenetic tree analysis of the HolP2110-like holins and several holins previously reported in *Ralstonia* phage, such as P2137, P2106 (their functions have been identified by our laboratory, although they have not been reported yet), RpY1 [26], Vb_RsoP_BMB50 [27], and phiAp1 were isolated [28] (Table 2). The results demonstrated that the HolP2110-like holins cluster separately from the reported *Ralstonia* phage holins (Figure 5), and the sequence alignment revealed very low similarity. Therefore, we proposed classifying the *Ralstonia* phages containing HolP2110-like holins into a new group, tentatively named the sigholinviridae group.

### 2.4. Impact of LysP2110 and HolP2110 Expression on Bacterial Growth

To investigate the mechanism of phage P2110 lysis, we constructed recombinant cell Pet28b-Lys and recombinant cell Pet28b-Hol to express LysP2110 and HolP2110, respectively. The impact of the induced recombinant protein expression on bacterial growth was observed. Western blotting analysis showed that the size of the expressed protein was consistent with the predicted size, indicating successful recombinant expression of LysP2110 and HolP2110 in recombinant cells (Figure 6A). Furthermore, a single band was detected in the isolated membranes. This result confirmed that HolP2110 is a membrane protein. Within 140 min of induction, the expression of LysP2110 did not affect bacterial growth (Figure 6B), but the expression of HolinP2110 led to the formation of nonspecific pores in the bacterial membrane, which inhibited bacterial growth (Figure 6B). The lack of effect from LysP2110 expression may be due to the absence of the holin protein, which prevents transmembrane translocation and contact with the peptidoglycan layer.

Consistent with the antibacterial activity test results, the amount of released β-galactosidase from recombinant cell Pet28b-Lys and recombinant cell Pet28b-Hol was detected after 140 min of induction. The expression of HolP2110 destabilized the integrity of the bacterial cell membrane, resulting in a large amount of β-galactosidase being detected outside the cell (Figure 7A), and a noticeable color change was observed (Figure 7C). In contrast, LysP2110 expression did not damage the cell membrane, and no significant color change was observed compared to the control group (Figure 7C). The concentration of extracellular nucleic acid was also measured after induction to further demonstrate the damage to the cell membrane structure. Compared with recombinant cell Pet28b-Lys, recombinant cell Pet28b-Hol had a higher concentration of extracellular DNA/RNA (Figure 7B), indicating membrane structure damage and cytoplasm release. In contrast, there was no significant difference between the recombinant cell Pet28b-Lys and the control group (Figure 7B), indicating that the cell membrane structure remained intact.

### 2.5. LysP2110 and HolP2110 Co-Expression and Their Effects on Cells

To further investigate the interplay between LysP2110 and HolP2110 in phage P2110 lysis, we cloned the genes into the co-expression vector PetDuet1 and induced expression in *Escherichia coli* BL21 (DE3). When expressed separately, neither LysP2110 nor HolP2110 caused a significant reduction in bacterial numbers. However, when co-expressed, the effect was dramatic: bacterial growth was severely inhibited (Figure 8A), and the culture became clear and showed significant amounts of cell debris. After 140 min of induction, the active cell counts showed that co-expression of LysP2110 and HolP2110 reduced the bacterial numbers by 92.1% compared to the control group (Figure 8B). These results indicated that LysP2110 and HolP2110 act synergistically in phage P2110 lysis of bacterial cells.

We evaluated the effect of co-expression on the bacterial cell membrane using the β-galactosidase assay (Figure 9A). As with the individual expression of each gene using the Pet28b vector, there was no significant color change in the empty PetDuet1 expression vector or the LysP2110-expressing group, indicating that the cell membrane remained intact and β-galactosidase in the cells could not be released into the extracellular environment, and bacterial growth was not inhibited (Figure 9B). However, when HolP2110 was induced alone, a significant color change was observed (Figure 9B), indicating damage to the cell membrane. Co-expression of LysP2110 and HolP2110 resulted in an even more profound color change than the control group (Figure 9B), indicating that the bacterial cell membrane was severely damaged, releasing a large amount of β-galactosidase and extensive bacterial lysis.

To further support these findings, the extracellular nucleic acid concentrations were measured. Co-expression of LysP2110 and HolP2110 resulted in a significantly higher concentration of DNA/RNA in the extracellular environment (Figure 10A,B), indicating extensive leakage of the intracellular components due to severe damage to the cell membrane. This was in contrast to the individual induction of HolP2110 and LysP2110, where a much lower level of extracellular DNA/RNA was detected. The results suggested that the synergistic effect of both proteins was responsible for the more profound damage to the cell membrane, resulting in extensive bacterial lysis.

### 2.6. Purified LysP2110 Has Broad-Spectrum Antibacterial Activity

To investigate whether the endolysin LysP2110 is capable of lysing bacteria in the extracellular environment and to determine its host range, we cloned the gene encoding LysP2110 from phage P2110 into the Pet28b vector. We expressed the recombinant protein with a histidine tag, which was subsequently purified using Ni–NTA agarose resin. SDS-PAGE analysis confirmed the presence of a single protein band at 25.9 kDa, matching the predicted size (Figure 11A).

Preliminary experiments revealed that the direct addition of extracellular LysP2110 had no effect on the growth of the Gram-negative strain Tb1546 (*Ralstonia solanacearum*), which is consistent with the bioinformatics analysis, indicating that LysP2110 lacks a transmembrane domain or signal peptide and cannot undergo transmembrane translocation from either the inside or outside of the cell [29]. Instead, it binds to peptidoglycan to achieve lysis. Previous studies have shown that adding endolysins directly to the extracellular environment can lead to lysis in Gram-positive bacteria, which lack an outer membrane structure and are thus susceptible to endolysin attack on peptidoglycan. However, the cell membrane of Gram-negative bacteria is composed of a three-layer structure, including the inner membrane, peptidoglycan layer, and outer membrane (lipoprotein, lipid bilayer, and lipopolysaccharide) [30]. In the bacterial cytoplasm, endolysin can achieve lysis with the assistance of the holin protein. According to Briers [31], ethylenediaminetetraacetic acid (EDTA) can remove divalent cations from the binding sites on the outer membrane, resulting in outer membrane damage. Adding an outer membrane permeabilizer such as EDTA (2 mM) can help endolysin effectively bind to the peptidoglycan targets through the bacterial outer membrane.

Our study showed that there was no significant change in the number of bacteria in the negative control group, regardless of the incubation time, indicating that EDTA (2 mM) did not affect Tb1546 (*Ralstonia solanacearum*) growth. Under the same conditions, the number of Tb1546 treated with LysP2110 decreased by 50% within 80 min, while the positive control group, treated with lysozyme, decreased by 75%. These findings demonstrated the strong antibacterial activity and effectiveness of LysP2110 (Figure 11B).

Additionally, we selected eight Gram-negative bacteria and one Gram-positive bacterium to determine the lysis spectrum of LysP2110. Following treatment with LysP2110 at a concentration of 0.2 mg/mL, the antibacterial activity was measured, and the result showed that the antibacterial activity was 95.0%, 96.9%, 90.3%, 96.6%, 28.3%, 86.6%, 86.3%, 74.7%, and 43.5%, respectively. The results demonstrated that LysP2110 exhibits highly efficient bactericidal activity against the host bacteria (*Ralstonia solanacearum*), with the antibacterial activity exceeding 70% in most cases. Furthermore, it also showed significant bactericidal efficacy against the other three Gram-negative bacteria, which are commonly associated with plant pathogens and contribute to plant diseases. However, its efficacy against *Ralstonia solanacearum*, isolated from eucalyptus potato (Po) and *Bacillus subtilis*, was relatively weak (Figure 12). It is possible that these bacterial strains exhibit potential resistance to the lysozyme, which warrants further investigation in future studies.

### 2.7. The N-Terminal Signal Peptide (SP) Plays an Important Role in HolP2110 Lysis

Since the signal peptide is located at the amino acid sites 1–30 of HolP2110, we constructed Hol_ΔSP_P2110 by deleting the 1–30 transmembrane structural domain and retaining the transmembrane structural domain at amino acids 39–61. Western blotting analysis showed that the size of the expressed protein was consistent with the predicted size (Lys: 22.2KDa, Hol_ΔSP_: 8.8 KDa), indicating the successful recombinant expression of LysP2110 and Hol_ΔSP_P2110. The recombinant cell (PetDuet-Hol_ΔSP_ + Lys) expression of Hol_ΔSP_P2110 did not appear to be toxic to host cells, in contrast to the results of the full-length HolP2110 expression. As previously described, when co-expressing HolP2110 with LysP2110, OD_600_ decreased by nearly two-thirds after 140 min of induction. In contrast, increased OD _600_ values in the presence of the SP deletion mutant indicated normal cell growth with little difference compared to the control, and the cell membrane remained intact. These findings suggest that the N-terminal SP of HolP2110 plays a key role in the localization of the holin protein to the inner cell membrane (Figure 13).

### 2.8. Sodium Azide Treatment Accelerates Bacterial Lysis Time

The addition of sodium azide and the resulting occurrence of premature bacterial lysis further confirm that the lysis system of phage P2110 is a two-component system of endolysin and holin [11]. Sodium azide blocks cell respiration and inhibits membrane H-ATPase, reducing the cellular ATP levels without affecting membrane potential [32]. In this study, sodium azide was added to the recombinant cell PetDuet1-Hol + Lys at concentrations of 0 mM, 1 mM, and 10 mM, and an empty vector (PetDuet1) *E. coli* was used as a control, induced with 1 mM IPTG. The bacterial growth was monitored.

The results indicate that the bacterial lysis time advances continuously as the concentration of sodium azide increases (Figure 14(a,b,c)). At a concentration of 10 mM, the bacterial growth showed immediate lysis. The occurrence of premature lysis suggests that there are enough holins and endolysins in the cell to achieve membrane perforation and peptidoglycan lysis under IPTG induction [33]. The holin protein determines the precise time for phage to dissolve bacteria, and its signal may be the depletion of intracellular ATP or depolarization of the membrane [33,34]. The co-expression recombinant system used in this study mimics a typical phage infection. During a normal phage infection, the decrease in intracellular ATP levels may be due to the synthesis and assembly of large virus molecules.

*Gaidelytė* et al. demonstrated through their early lysis experiments of *Bacillus subtilis* cells infected with phage Bam35 and induced by arsenate that phages regulate the lysis time by monitoring the ATP levels [35]. Furthermore, sodium azide (NaN_3_) has been widely used as an inhibitor of the ATPase activity of SecA to detect whether the Sec system is involved in the secretion process [36]. Therefore, based on the above experimental results, it is hypothesized that the depletion of intracellular ATP may serve as the initiation signal for phage P2110 lysis. Previous bioinformatics analysis of HolP2110 suggested that the signal of decreased ATP levels is transmitted to the holin protein through the cell’s Sec protein secretion system. HolP2110 is predicted to have a signal peptide structure of a Sec secretion system, which could be secreted by the bacterial Sec system and integrated into the cell membrane [37]. When intracellular ATP levels decrease, the secreted holins may not localize to the membrane, leading to depolarization of the membrane and the formation of large nonspecific pores, which then release endolysins inside the cell to facilitate bacterial lysis. However, these hypotheses require further research to confirm.

### 2.9. Early Lysis Leads to the Formation of Smaller Pores by Holin Proteins on the Inner Membrane

Based on the previous experiments, a further investigation was conducted to explore how holin affects the initiation time of lysis by decreasing the ATP levels. We constructed a recombinant cell (Pet28b-Hol) and established three treatment conditions. The experimental group (10 mM) was induced for holin expression with 1 mM IPTG in the presence of 10 mM sodium azide inhibitor. The negative control group (control) was treated with 10 mM sodium azide without IPTG induction for holin expression, while the positive control group (0 mM) was induced with 1mM IPTG for holin expression without the sodium azide treatment (Figure 15). Bacterial growth and cell membrane permeability were observed.

The experimental results indicated that recombinant cell growth is inhibited or ruptured when the holin forms a pore on the inner membrane to trigger the lysis program. The results showed that under the influence of the sodium azide inhibitor, the time for holin to initiate the lysis program was earlier than the positive control group (Figure 15A), which is consistent with the results of co-expression of LysP2110 and HolP2110. This confirmed that the decrease in the ATP level controls the overall bacterial lysis time by affecting the holin.

To further investigate the specific regulatory mechanism of HolP2110, extracellular β-galactosidase activity was measured at each time point to predict cell membrane permeability. The negative controls did not detect extracellular β-galactosidase within 100 min, indicating that the sodium azide treatment did not damage the cell membrane (Figure 15B). The positive controls showed a rapid increase in extracellular β-galactosidase content (Figure 15B) 40 min after triggering lysis with the induction treatment (Figure 15A(a)). Interestingly, the experimental group failed to detect β-galactosidase in the extracellular fluid 40 min after induction. However, based on the results of the bacterial growth experiment, the experimental group had already started to trigger lysis 20 min after induction (Figure 15A(b)). Moreover, after 100 min of induction treatment, the amount of β-galactosidase detected in the extracellular fluid of the positive control group was much higher than that of the experimental group (Figure 15C). This seems inconsistent with the results of the bacterial growth experiment, which indicate that the permeability of the cell membrane is not high.

Therefore, it is hypothesized that the decrease in the ATP level leads to the formation of smaller pores by holin proteins on the inner membrane. Since the internal lysin LysP2110 is smaller, with a molecular weight of 22.2 KDa, the inhibitor has no significant effect on the lysed cells co-expressing LysP2110 and HolinP2110. However, β-galactosidase is a dimeric protein with a molecular weight of 154 KDa, which cannot pass through the small pore formed by holin, resulting in a much lower extracellular content compared to the control group. This is consistent with Wang et al.’s [33] suggestion that energy toxin-induced early lysis will lead to the formation of smaller lesions by holin on the inner membrane.

## 3. Discussion

For the bacteriophage P2110, the endolysin LysP2110 and the holin HolP2110 are required for bacterial lysis, and they work in concert. HolP2110 regulates the timing of lysis initiation and creates a large nonspecific pore (The smallest diameter can pass through proteins with a molecular weight of 154 KDa) in the inner membrane. Subsequently, LysP2110 is released into the periplasmic space, where it binds to peptidoglycan and cleaves the β-(1,4) glycosidic bond in the peptidoglycan backbone. This disruption of the peptidoglycan layer leads to rapid lysis of the cell and the release of progeny phages due to the difference in osmotic pressure between the cell and the environment. Moreover, a series of experiments investigating the impact of the sodium azide energy inhibitor on the P2110 lysis system indicates that the decrease in intracellular ATP levels serves as a signal to induce bacteriophage-mediated bacterial lysis and is regulated through the unique structure of HolP2110. These findings hold promise for improving bacteriophage-based therapies in the future. For instance, one potential approach is to modify bacteriophages to delay lysis within bacterial cells, thereby maximizing the proliferation of phage progeny and achieving more efficient bacterial eradication.

Purified endolysin LysP2110 also demonstrates excellent antibacterial activity and broad-spectrum effects in the extracellular environment, making it a promising candidate for the future treatment of bacterial diseases. However, the current antibacterial activity of LysP2110 still relies on the assistance of outer membrane permeabilizers, and many of these permeabilizers are not suitable for medical, food, and industrial applications. Recent research has shown that fusion endolysins can be created through the artificial modification of endolysin structures, allowing for the lysis of Gram-negative bacteria. Engineered enzymes such as “artilysins” have been shown to have antibacterial effects on Gram-negative bacteria [38,39].

In addition, several studies have reported the application of lysozymes in the control of plant diseases [40,41,42]. The results demonstrate that lysozymes are capable of efficiently killing pathogenic bacteria without exerting toxicity on plants. In our future studies, we plan to artificially modify the structure of LysP2110 to enable it to achieve extracellular lysis without outer membrane permeabilizers. Furthermore, in order to effectively apply it in the treatment of bacterial wilting diseases, we intend to introduce the modified LysP2110 into *Bacillus subtilis* for its exogenous expression in the natural environment. The resulting biocontrol agent will be extensively deployed in plants infected with bacterial wilt for therapeutic purposes.

Wang et al. [33] proposed time- and function-based models of holins on the cell membrane. During the late stages of phage infection, holins and endolysins are expressed. Holins initially form homologous dimers in the inner membrane, which then accumulate in higher-order oligomers to form raft-like aggregates. (Figure 16) The formation of these aggregates allows the lipids inside to be excluded as much as possible. In the initial stages of aggregate formation, stability is maintained by lipid–protein interactions. As the concentration of holins increases in the inner membrane, the size of the aggregates continues to grow, and their stability becomes dominated by protein–protein interactions, which are facilitated by the alternating arrangement of peptide chains to interact with one another [15]. At some point in the accumulation process, thermal fluctuations in the protein aggregates create a transient channel or pore, causing intracellular K^+^ efflux and depolarization of the inner membrane. When the proton motive force (PMF) decreases by approximately 40%, the raft-like aggregates will form a large pore in the inner membrane, releasing endolysin to achieve bacterial lysis [43].

As HolP2110 is a structure unique to the *Caudoviricetes Ralstonia* phages, a series of experiments were carried out to explore whether there is a specific cleavage mechanism. Based on the analysis of the models from previous studies and the experimental results of this study, we propose a conjecture for the time and functional model of the phage P2110 lysis system. During the late stage of phage P2110 infection, the transcription and translation of the holin protein generates a precursor protein, which is then transported to the membrane via the Sec transport system in the cytoplasm through its signal peptide (Figure 17A). The enzyme system on the membrane facilitates the holin protein’s subsequent modification and folding. As the holin protein matures, the signal peptide region is cleaved and removed by membrane proteinases (IMPs) or other related enzymes on the membrane, releasing the mature holin protein. At the same time, the holin protein’s transmembrane region is inserted into the cell membrane, which allows it to locate and fixate onto the membrane, accumulating in a raft-like aggregate (Figure 17B). Since the synthesis of virus macromolecules and the assembly of progeny phages by the phage consume intracellular ATP, when the intracellular ATP level drops significantly, the expressed holin protein cannot be inserted into the membrane through the Sec secretion system. As the concentration of holin protein on the membrane no longer increases, the stability of the raft-like aggregate cannot be maintained by enhancing protein–protein interactions, and eventually, the proton motive force collapses under depolarization. The holin aggregate then forms a large nonspecific pore at the membrane, allowing endolysin to escape to the periplasm and achieve bacterial lysis (Figure 17C). However, further research is necessary to confirm these possibilities.

## 4. Materials and Methods

### 4.1. Bacterial Strains, Bacteriophages, Plasmids, and Growth Conditions

The bacterial strains, bacteriophages, and plasmids used in this study are listed in Table 3. *E. coli* BL21 (DE3) was purchased from Tsingke Biotechnology Co., Ltd. (Beijing, China), while other bacterial strains were obtained from the Bacterial Laboratory of South China Agricultural University. All strains were grown on the NA medium (0.5 g/L yeast extract, 5 g/L tryptone, 3 g/L beef extract, 10 g/L glucose, 15 g/L agar, pH 7.0) at 28 °C, except for *E. coli,* which was grown on the LB medium (10 g/L NaCl, 10 g/L tryptone, 5 g/L yeast extract, 15 g/L agar) at 37 °C for cloning and recombinant protein expression. The final concentrations of kanamycin (Km) and ampicillin (Amp) in the culture medium were 50 μg/mL and 100 μg/mL, respectively. Isopropyl-β-D-thiogalactopyranoside (IPTG) was used at a final concentration of 1 mM to induce protein expression under the control of the T7 promoter in *E. coli*. The Pet28b and PetDuet1 vectors were used for recombinant protein expression.

### 4.2. Genome Sequencing and Phylogenetic Analysis of Bacteriophage P2110

In this experiment, the genomic DNA of bacteriophage P2110 was extracted using the method developed by Jakociune [44], Soleimani-Delfan [45], and others. The genome sequencing was performed by Shanghai Lingyun Biotechnology Co., Ltd. (Shanghai, China, NCBI accession number OP947226.1). The GeneMarkS (http://topaz.gatech.edu/GeneMark/ (accessed on 4 March 2023)) was used to predict the coding genes in the bacteriophage P2110 genome. The predicted protein sequences were compared using NR, Swiss-Prot, eggNOG, KEGG, GO, and other databases to obtain the annotation information for the predicted genes. The phylogenetic tree of phageP2110 was constructed using the maximum likelihood method and LG substitution model in MEGA11, based on the terminase large subunit (TERL) sequences. The phylogenetic tree of HolP2110 was constructed using the neighbor-joining method and the Jones–Taylor–Thornton (JTT) substitution model in MEGA11.

### 4.3. Prediction and Bioinformatics Analysis of the Lysis System in the Complete Genome of Bacteriophage P2110

The homologous proteins of endolysin LysP2110 were compared using blastp in NCBI, and the conserved domain was analyzed using the NCBI Conserved Domain Search (https://www.nih.gov/ (accessed on 5 March 2023)). TMHMM 2.0 (TMHMM 2.0, DTU Health Tech, Bioinformatic Services), SignalP 5.0 (SignalP 5.0, DTU Health Tech, Bioinformatic Services), and CCTOP (https://cctop.ttk.hu/ (accessed on 5 March 2023)) were used to predict the TMHD and signal peptides. The isoelectric point and molecular weight of proteins were predicted using Expasy (Expasy—Compute pI/Mw tool), and the protein solubility and secondary structure were predicted using Expasy (Expasy—ProtScale). The promoter was predicted by using Bprom (http://www.softberry.com/ (accessed on 5 March 2023)). The three-dimensional structure models of the LysP2110 and HolP2110 proteins were predicted using AlphaFold2 (https://colab.research.google.com/github/sokrypton/ColabFold/blob/main/AlphaFold2.ipynb (accessed on 4 March 2023)).

### 4.4. Construction of Recombinant Plasmids and Cells

The gene fragment of LysP2110 was amplified by PCR using the bacteriophage P2110 genomic DNA and primers 28b-Lys-F/R (Table 4). The gene fragment was then cloned into the Pet28b vector using the restriction enzymes EcorI, HindIII, and T4 ligase. The same method was used to construct the Pet28b-Hol, PetDuet-Lys, PetDuet-Hol, PetDuet-Hol_ΔSP _+ Lys, and PetDuet-Hol + Lys vectors. Finally, the recombinant plasmid was transformed into *Escherichia coli* BL21 (ED3) by heat-shock transformation for 60 s. The positive recombinant cell was confirmed by PCR and sequencing. The identified recombinant cell was named recombinant cell Pet28b-Lys. The term “recombinant cells” mentioned in the text refers to the *Escherichia coli* BL21 (DE3) cells carrying the aforementioned recombinant plasmid.

### 4.5. Growth Measurement

The effect of expressing recombinant proteins on bacterial growth was analyzed by measuring the optical density (OD) at 600 nm using a spectrophotometer. The recombinant cells were cultured to the logarithmic growth phase (OD_600_ = 0.6–0.8) and induced with IPTG (50 mg/mL) to a final concentration of 1 mM. The cultures were incubated at 37 °C and 200 rpm for 140 min. The OD values were measured every 20 min using the LB medium as a control. The experiment was repeated three times.

### 4.6. Detection of Extracellular β-Galactosidase Activity and Nucleic Acid Concentration

Recombinant cells were cultured to the logarithmic growth phase (OD_600_ = 0.6–0.8), and IPTG (50 mg/mL) was added to the bacterial suspension at a final concentration of 1 mM for recombinant protein induction. The culture was incubated at 37 °C and 200 rpm for 140 min. The induced bacterial suspension was collected and centrifuged at 12,000 rpm for 5 min, and then 500 μL of extracellular supernatant and 100 μL of o-nitrophenyl-β-galactoside (ONPG) (20 mM) were added. The mixture was incubated in a 45 °C water bath for 30 min. The activity of β-galactosidase was detected by measuring the absorbance at 420 nm (OD_420_) using a spectrophotometer. The detection of the extracellular nucleic acid concentration was performed similarly. After inducing the bacterial suspension, the extracellular supernatant was collected, and the nucleic acid concentration in the supernatant was determined using a Nandrop instrument. The excretion of intracellular nucleic acid was analyzed.

### 4.7. Purification of LysP2110, Antibacterial Activity, Determination of Lysis Spectrum, and Western Blotting

Recombinant cells (pet28b-Lys) were cultured in the LB medium (containing 50 μg/mL kanamycin) to the logarithmic growth phase (OD_600_ = 0.6–0.8). IPTG was added to the medium at a final concentration of 1 mM. The culture was incubated at 37 °C and 200 rpm/min for 4 h. The bacterial precipitate was collected by centrifugation, resuspended in lysis buffer (0.3 M NaCl, 50 mM Na_2_HPO_4_, 10 mM PMSF, pH 8), and sonicated (200 W, 2 s on and 4 s off) until the suspension became clear. The supernatant was obtained by centrifugation at 10,000× *g* for 30 min at 4 °C. The target protein in the supernatant was purified using Ni–NTA resin (BioSun Biochemicals (Beijing, China)), according to the instructions, to obtain the crude LysP2110 enzyme. The purified protein was analyzed by SDS-PAGE. Western blotting was conducted using an Anti-His-tagged antibody following the reported protocols [46]. The protein expression analysis of the recombinant cell Pet28b-Hol and recombinant cell PetDuet-Hol_ΔSP_ + Lys was performed using the same method as described above.

The antibacterial activity of LysP2110 against Tb1546 (*Ralstonia solanacearum*) was determined. Tb1546 bacterial suspension in the logarithmic growth phase (OD_600_ = 0.8) was collected by centrifugation at 5000 rpm for 5 min and washed with PBS (pH 7.4) containing 20 mM EDTA. Then, 100 μL of the bacterial suspension was mixed with 100 μL of the crude LysP2110 enzyme (0.6 mg/mL) and incubated at 28°C and 200 rpm for 80 min. A positive control was set up by adding an equal amount of lysostaphin (BioSun Biochemicals), and 1 × PBS was used as a negative control. The absorbance at 600 nm was measured every 20 min using a spectrophotometer. The experiment was repeated three times.

Determination of the cleavage spectrum of LysP2110. Eight strains of Gram-negative bacteria and Gram-positive bacteria (*Bacillus subtilis*) were selected from those preserved in our laboratory (Table 3). Similar to the aforementioned experimental steps, bacterial cultures were grown to the logarithmic growth phase (OD_600_ = 0.6), and the bacterial precipitation was obtained by centrifugation at 5000 rpm for 5 min. The bacterial precipitation was washed three times with an equal volume of PBS and then diluted to 10^5^ CFUs/mL with PBS (containing 20 mM EDTA). An amount of 100 μL of the bacterial suspension was mixed with 100 μL of the LysP2110 crude enzyme solution (0.2 mg/mL) and incubated at 28 °C and 200 rpm for 30 min. An equal volume of 1 × PBS was added as a blank control. After incubation, 200 μL of the bacterial culture was evenly spread on the LB solid medium after incubation at 28 °C for 10 h. The colony count was observed, and the antibacterial activity was calculated using the following formula. The experiment was repeated three times.
$$\mathrm{Antibacterial}\;\mathrm{activity}\left(\%\right)=\frac{\mathrm{CFUs}\left(\mathrm{CK}\right)-\mathrm{CFUs}\left(\mathrm{LYS}\right)}{\mathrm{CFUs}\left(\mathrm{CK}\right)}\times 100$$

### 4.8. Effect of Energy Inhibitors on the PhageP2110 Lysis System

Following the experimental method described by Briers [47], the recombinant cell Pet28b-Hol and recombinant cell PetDuet-Hol + Lys were cultured to the logarithmic growth phase (OD_600_ = 0.6–0.8), and IPTG (50 mg/mL) was added to the bacterial culture to a final concentration of 1mM to induce the expression of the recombinant protein. Sodium azide was added to the bacterial culture at final concentrations of 0 mM, 1 mM, and 10 mM, respectively, and the bacterial culture was incubated at 37 °C and 200 rpm. The LB medium without bacteria was used as a control. The OD value of the induced bacterial suspension was measured every 20 min, and the experiment was repeated three times.

### 4.9. Statistical Analysis

Analysis of variance (ANOVA) using GraphPad Prism (version 8.0.1) was employed to assess the statistical differences. The significance level was set at *p* < 0.05. The ANOVA analysis allowed us to determine the significant differences among the main treatments and their interactions. Prior to conducting the ANOVA, we performed tests to check for normality and homogeneity of variances.

## 5. Conclusions

This study identified and analyzed the lysin–holin lysis system of bacteriophage P2110, which effectively kills bacterial growth in *Ralstonia solanacearum*. LysP2110 lacks a transmembrane domain and signal peptide and has no antibacterial activity when used alone. It requires HolP2110 to achieve transmembrane translocation and lyse the bacterial peptidoglycan layer. The bioinformatics analysis showed that the HolP2110 protein structure is unique to the *Ralstonia solanacearum* phages, with the transmembrane domain of amino acids 39–61 and a signal peptide structure of amino acids 1–30. We also explored the regulatory mechanism of the P2110 phage holin protein on lysis time and proposed a lysis model hypothesis.

Moreover, LysP2110 was purified and tested for antibacterial activity, showing broad-spectrum activity with varying degrees of bactericidal activity against three Gram-negative bacteria and one Gram-positive bacterium in addition to its host, *Ralstonia solanacearum*. Our next step is to perform artificial modifications to the protein structure of LysP2110, aiming to enable its penetration through the outer membrane of *Ralstonia solanacearum* in the natural environment and achieve highly efficient antibacterial effects. In the future, LysP2110 can be developed as a bacterial inhibitor, addressing the challenging issue of difficult treatment for plant bacterial wilt disease while reducing antibiotic resistance and pollution. Thanks to its broad-spectrum antimicrobial activity, it can even be explored as a therapeutic agent for other plant bacterial diseases.

## Figures and Tables

**Figure 1 ijms-24-10375-f001:**
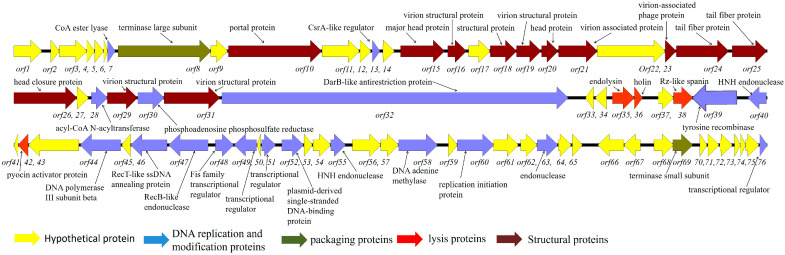
Genome map of phage P2110. Block arrows indicate open reading frames (ORFs) and the direction of each read. Each ORF is numbered in running order, as indicated.

**Figure 2 ijms-24-10375-f002:**
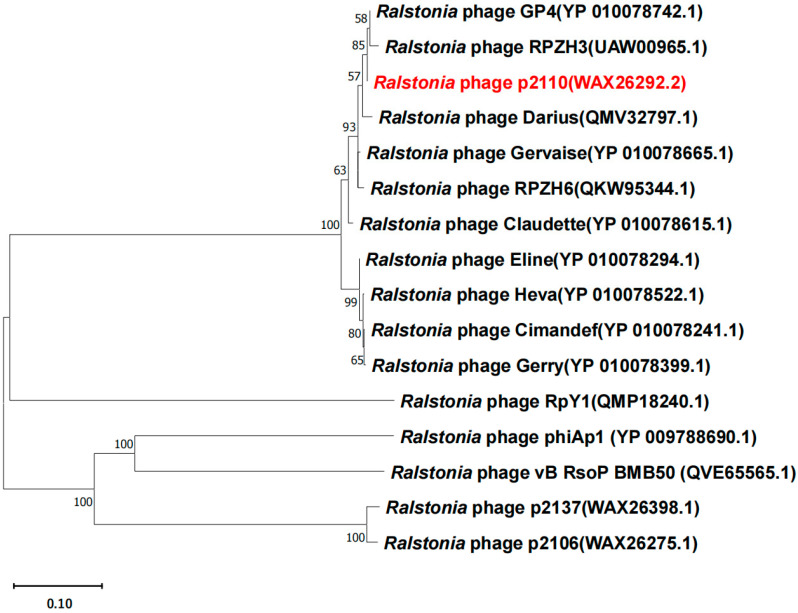
The evolutionary relationships between P2110 phage and related phage species were observed based on terminase large subunit (TERL) sequences. The evolutionary history was inferred using the maximum likelihood method and LG substitution model in MEGA11. The tree is drawn to scale, with branch lengths in the same units as those of the evolutionary distances used to infer the phylogenetic tree. Bootstrap values (expressed as percentages of 1000 replications) are shown at branching points. GenBank accession numbers are shown in parentheses.

**Figure 3 ijms-24-10375-f003:**
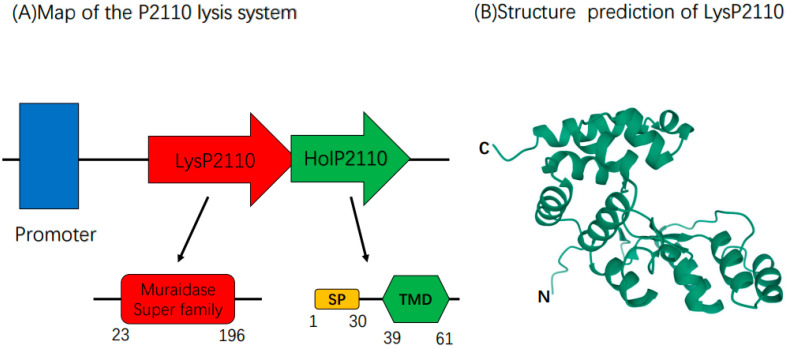
(**A**) Schematic representation of the P2110 lysis system. Numbers indicate the amino acid position. SP: Signal peptide. TMD: transmembrane domain. (**B**) Three-dimensional structure prediction of LysP2110. The pLDDT = 93.9. The pTM = 0.84.

**Figure 4 ijms-24-10375-f004:**
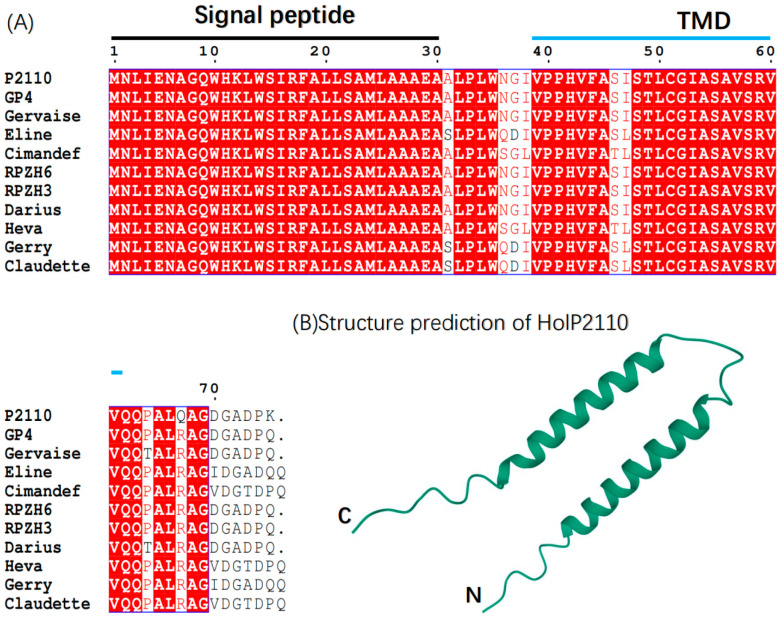
(**A**) Sequence alignment of HolP2110 with that of phage Gervaise, GP4, Eline, Cimandef, RPZH6, RPZH3, Darius, Heva, Gerry, and Claudette (Accession No. (Table 2)). Amino acids 1–30 are the putative signal peptide, and amino acids 39–61 are the putative transmembrane domain. (**B**) Three-dimensional structure prediction of HolP2110. The pLDDT = 58.9. The pTM = 0.346.

**Figure 5 ijms-24-10375-f005:**
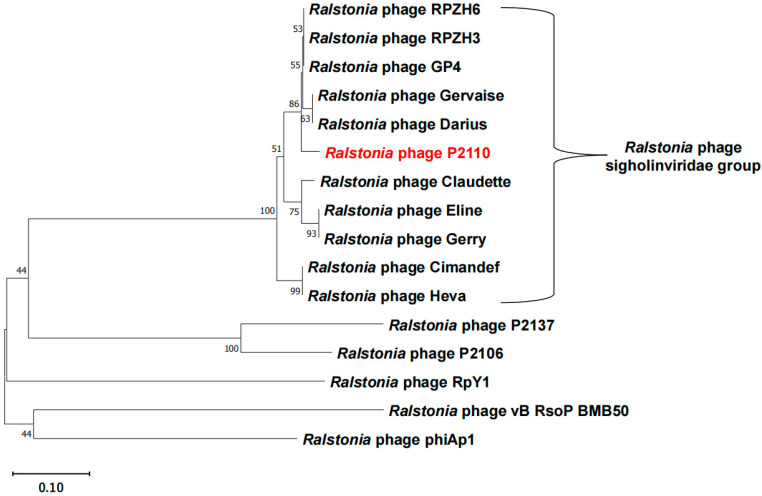
The evolutionary relationships between HolP2110 and related phage holins. The evolutionary history was inferred using the neighbor-joining method and the Jones–Taylor–Thornton (JTT) substitution model in MEGA11. The tree is drawn to scale, with branch lengths in the same units as those of the evolutionary distances used to infer the phylogenetic tree. Bootstrap values (expressed as percentages of 1000 replications) are shown at branching points.

**Figure 6 ijms-24-10375-f006:**
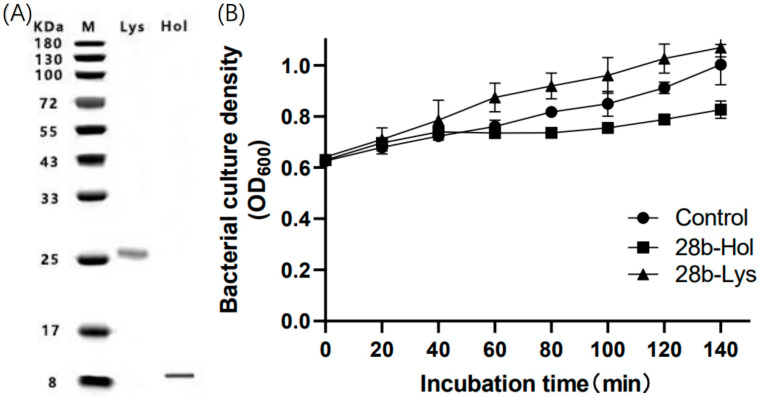
Expression and antibacterial activity of X2-Lys and X2-Hol. (**A**) LysP2110 and HolP2110 can be expressed in vitro. Lane M: marker. Lane Lys: LysP2110. Lane Hol: HolP2110. (**B**) Bacterial growth.

**Figure 7 ijms-24-10375-f007:**
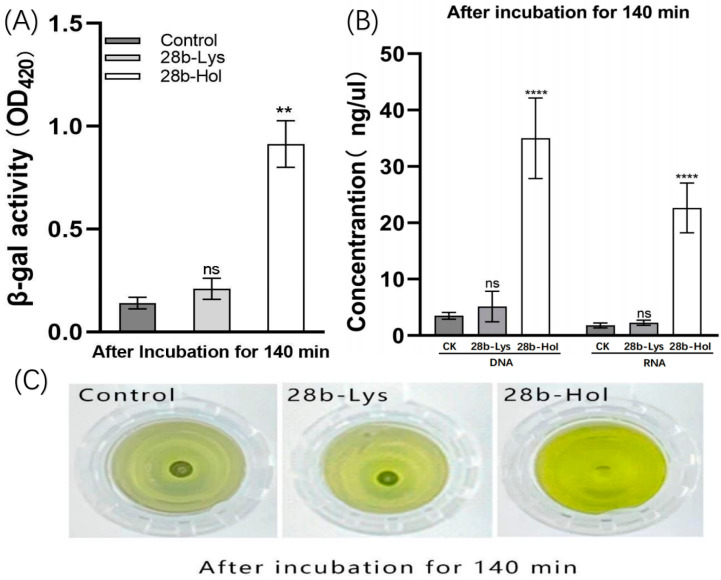
(**A**) β-galactosidase activity (OD420) after 140 min post-induction. Asterisks (**) above each bar represent significant differences (*p* < 0.05) in β-galactosidase activity among the three groups of control, 28a-Lys, and 28a-Hol at the same time point. (**B**) The release of DNA/RNA after 140 min post-induction. Asterisks (****) above the bar represent significant differences (*p* < 0.05) in RNA or DNA concentrations between control, 28a-Lys, and 28a-Hol groups. 28a-Lys/Hol: Pet28a carrying LysP2110/HolP2110. (**C**) Change in color of solutions in control, 28-Lys, and 28-Hol groups after 140 min incubation.

**Figure 8 ijms-24-10375-f008:**
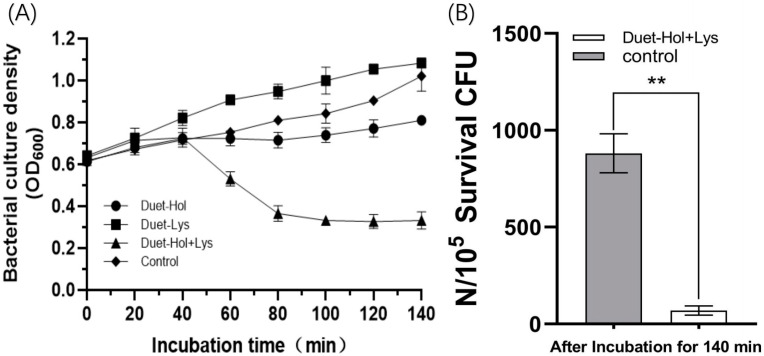
Co-expression of LysP2110 and HolP2110 and its effect on bacterial growth. (**A**) Bacterial growth. Duet-Lys/Hol: PetDuet1 carrying LysP2110/HolP2110. (**B**) Active bacteria count. Asterisks (**) above the bar represent significant differences (*p* < 0.05) in surviving bacteria between control and Duet-Hol + Lys groups.

**Figure 9 ijms-24-10375-f009:**
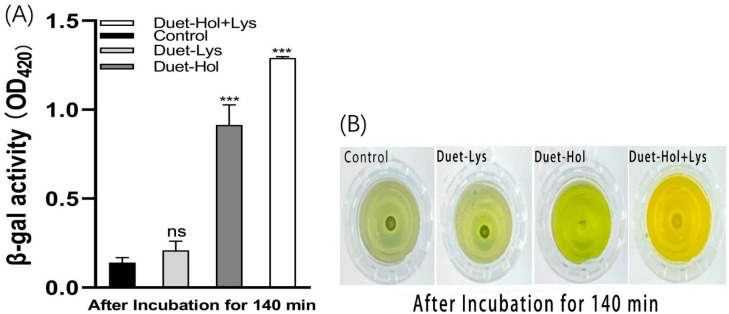
(**A**) β-galactosidase activity (OD_420_) after 140 min post-induction. Asterisks (***) above each bar represent significant differences (*p* < 0.05) in β-galactosidase action between the group and control, Duet-Lys, Duet-Hol, and Duet-Hol + Lys at the same time point. No significance (ns) above the bar represent no significance in β-galactosidase action between the group and control. (**B**) Change in color of solutions in control, Duet-Lys, Duet -Hol, and Duet-Hol + Lys groups after 140 min incubation.

**Figure 10 ijms-24-10375-f010:**
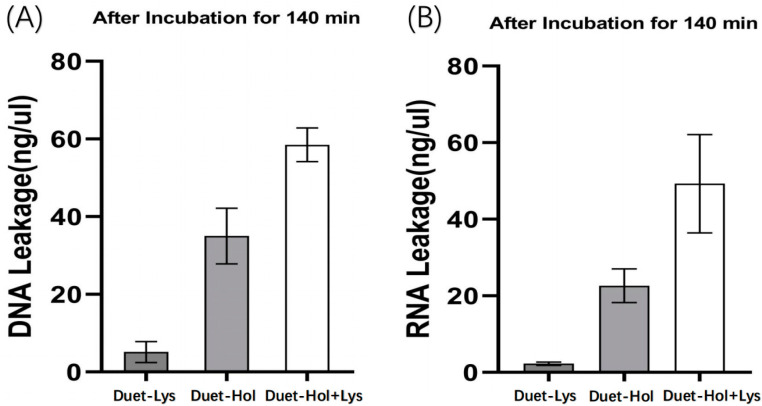
(**A**) The release of DNA among the three groups of Duet-Lys, Duet-Hol, and Duet-Hol + Lys after 140 min of incubation. (**B**) The release of RNA among the three groups of Duet-Lys, Duet-Hol, and Duet-Hol + Lys after 140 min of incubation.

**Figure 11 ijms-24-10375-f011:**
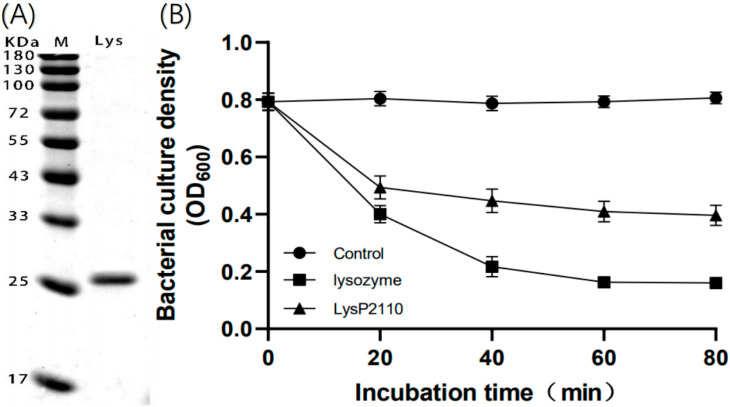
(**A**) SDS-PAGE of purified LysP2110. Lane M: marker. Lane Lys: purified LysP2110. (**B**) Antibacterial activity of LysP2110 with the help of EDTA. Control: negative control (PBS). Lysozyme: positive control (purchased from Beyoncé Biologicals (Shanghai, China)). Mean ± SD is shown (*n* = 3).

**Figure 12 ijms-24-10375-f012:**
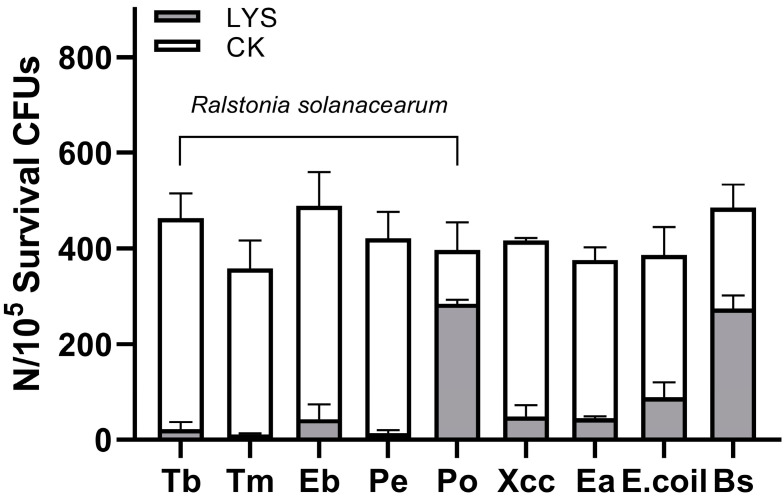
Determination of cleavage profile of bacteria by LysP2110. Tb, Tm, Eg, Pe, and Po are *Ralstonia solanacearum* isolated from tobacco, tomato, Egg plant, peanut, and potaoto, respectively. Xcc: *Xanthomonas citri* subsp. *citri.* Pcc: *Pectobacterium carotovorum* subsp. *carotovorum*. *E. Coil*: *Escherichia coli*. Bs: *Bacillus subtilis*.

**Figure 13 ijms-24-10375-f013:**
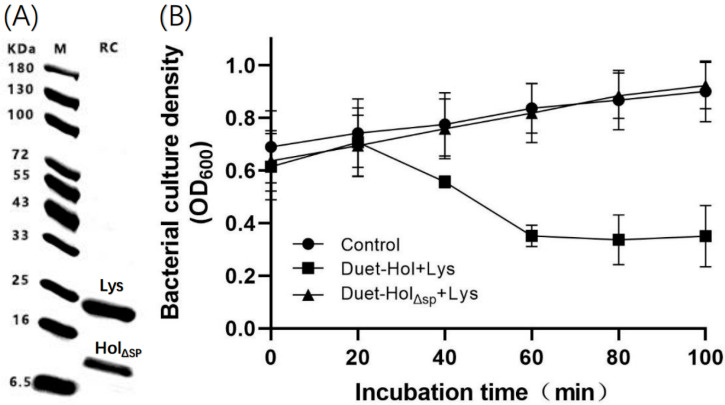
(**A**) LysP2110 and Hol_ΔSP_P2110 were expressed in the recombinant cell. Lane M: marker. Lane RC: recombinant cell. (**B**) Effect of HolP2110_ΔSP_ expression on recombinant cell (PetDuet-HolΔSP + Lys) growth.

**Figure 14 ijms-24-10375-f014:**
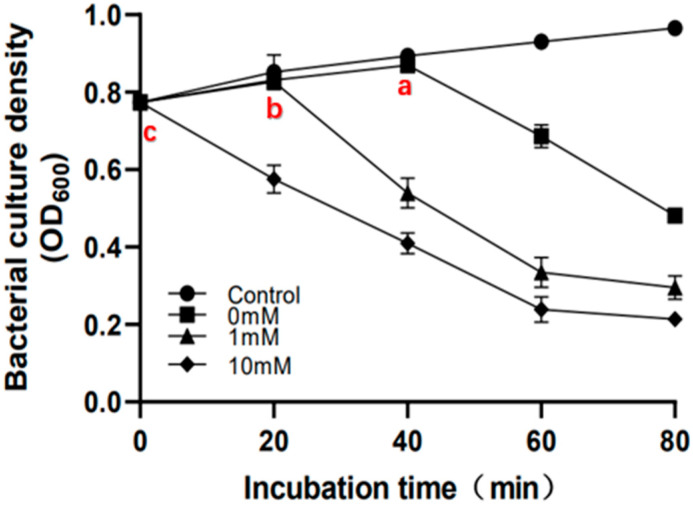
Effect of sodium azide on antibacterial activity of lysis systems. Control: Induced by 10 mM sodium azide treatment without the addition of IPTG. (a–c) The inflection points of bacterial OD_600_ values in response to different concentrations of sodium azide. (a) The inflection point for the bacterial OD_600_ value in the absence of sodium azide (0 mM). (b) The inflection point for the bacterial OD_600_ value in the absence of sodium azide (1 mM). (c) The inflection point for the bacterial OD_600_ value in the absence of sodium azide (10 mM).

**Figure 15 ijms-24-10375-f015:**
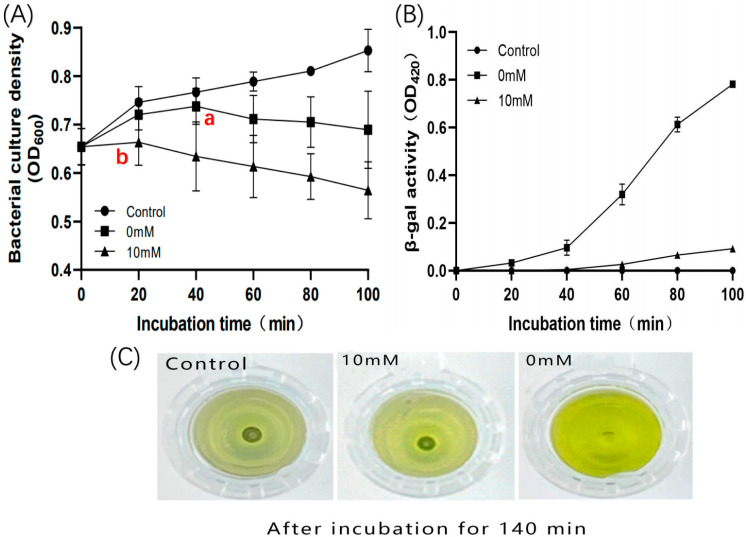
Effect of sodium azide on HolP2110. (**A**) Growth of the recombinant cell (Pet28b-Hol). Control: treated with 10 mM sodium azide without IPTG induction; 0 mM: induced with 1 mM IPTG without sodium azide treatment; 10 mM: induced with 1 mM IPTG in the presence of 10 mM sodium azide inhibitor. (a,b) Time to initiate cleavage. (**B**) β-galactosidase activity (OD_420_). (**C**) Change in color of solutions in control, 10mM, and 0mM groups after 140 min incubation.

**Figure 16 ijms-24-10375-f016:**
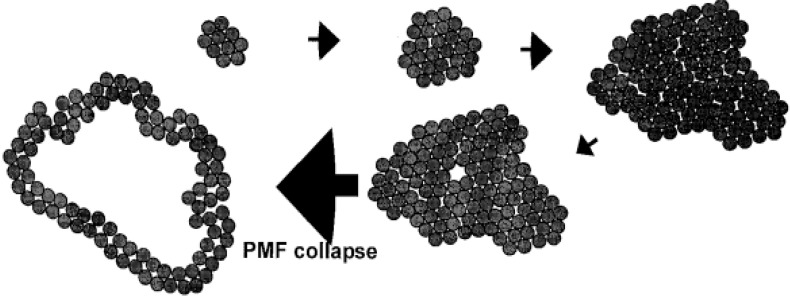
Time and functional models of holin on the cell membrane [33].

**Figure 17 ijms-24-10375-f017:**
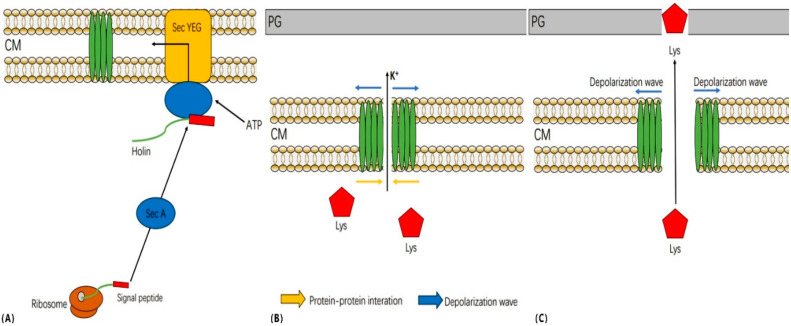
The model of the phage P2110 lysis system. (**A**) Holin expression and insertion at the endothelium via the sec secretion system. (**B**) Raft-like aggregates maintain a steady state before triggering. (**C**) Lysis trigger and lysozyme release. PG: peptidoglycan. CM: intracellular membrane.

**Table 1 ijms-24-10375-t001:** Comparison of the available genomes of *Ralstonia solanacearum* phages from the GeneBank database.

Name	Length	Accession No.
*Ralstonia* phage p2110	59,380 bp	OP947226.1
*Ralstonia* phage GP4	61,129 bp	NC_054964.1
*Ralstonia* phage Claudette	57,085 bp	NC_054962.1
*Ralstonia* phage Gervaise	61,164 bp	NC_054963.1
*Ralstonia* phage RpY1	43,284 bp	QMP18240.1
*Ralstonia* phage RPZH6	64,657 bp	MT361768.1
*Ralstonia* phage Darius	60,734 bp	MT740732.1
*Ralstonia* phage RPZH3	65,958 bp	MZ870514.1
*Ralstonia* phage Cimandef	55,171 bp	NC_054956.1
*Ralstonia* phage Heva	58,352 bp	NC_054960.1
*Ralstonia* phage P2137	37,595 bp	WAX26398.1
*Ralstonia* phage P2106	35,928 bp	WAX26275.1
*Ralstonia* phage Gerry	60,898 bp	NC_054959.1
*Ralstonia* phage Eline	57,352 bp	NC_054957.1
*Ralstonia* phage phiAp1	44,793 bp	YP 009788690.1
*Ralstonia* phage vB RsoP BMB50	43,665 bp	QVE65565.1

Data obtained from GeneBank.

**Table 2 ijms-24-10375-t002:** Comparison of the protein of HolP2110 from the GenBank database.

Name	Source	Length	Accession No.
holin	*Ralstonia* phage Cimandef	76	YP_010078203.1
holin	*Ralstonia* phage p2110	75	WAX26318.1
holin	*Ralstonia* phage GP4	75	YP_010078796.1
holin	*Ralstonia* phage Gervaise	75	YP_010078692.1
holin	*Ralstonia* phage Eline	76	YP_010078319.1
holin	*Ralstonia* phage Claudette	76	YP_010078659.1
holin	*Ralstonia* phage Heva	76	YP_010078482.1
hypothetical protein	*Ralstonia* phage Darius	75	QMV32770.1
hypothetical protein	*Ralstonia* phage Alix	76	QMV32463.1
hypothetical protein	*Ralstonia* phage RPZH3	75	UAW01020.1
hypothetical protein	*Ralstonia* phage RPZH6	75	QKW95398.1
hypothetical protein	*Ralstonia* phage P2137	70	WAX26396.1
hypothetical protein	*Ralstonia* phage P2106	69	WAX26273.1
holin	*Ralstonia* phage BMB50	60	QVE65567.1
holin	*Ralstonia* phage phiAp1	152	APU03191.1
holin	*Ralstonia* phage RpY1	72	QMP18260.1

Data obtained from GeneBank.

**Table 3 ijms-24-10375-t003:** Bacteria, phage, and plasmids used in this study.

Type	Organism	Name	Source
Bacteria	*Ralstonia solanacearum*	Tb1546	SCAUBL
		TmSZ-1	SCAUBL
		Eg1906	SCAUBL
		Po36	SCAUBL
		Pe1301	SCAUBL
	*Pectobacterium carotovorum* subsp. *caroto-vorum*	Pcc1806	SCAUBL
	*Xanthomonas citri* subsp. *citri*	Xcc63	SCAUBL
	*Bacillus subtilis*	Bs03	SCAUBL
	*Escherichia coli*	BL21(ED3)	BJTB
Phage	*Ralstonia* phage	P2110	SCAUBL
Plasmids	ND	Pet28b	BJTB
		PetDuet1	BJTB

ND: not determined. SCAUBL: South China Agricultural University, Bacterial Laboratory. BJTB: Beijing Tsingke Biotechnology.

**Table 4 ijms-24-10375-t004:** Primers used in this study.

Primer	Sequences (5′-3′)	Size (bp)	Characterization
28b-Lys-F	CCCCATATGATGCCACAAACTAGCAAGC	28	Gene of Lys from P2110 phage
28b-Lys-R	TTCTCGAGTCATGCGGGTCCCGTTTG	26	
28b-Hol-F	GAATTCATGAACCTCATCGAAAACGCC	27	Gene of Hol from P2110 phage
28b-Hol-R	AAGCTTGGTCTTCACTTCGGGTCGG	25	
Duet-Lys-F	CCCATATGCCACAAACTAGCAAGC	24	Gene of Lys from P2110 phage
Duet-Lys-R	TTCTCGAGTCATGCGGGTCCCGTTTG	26	
Duet-Hol-F	CCGAATTCGATGAACCTCATCGAAAACGCC	30	Gene of Hol from P2110 phage
Duet-Hol-R	TTAAGCTTGGTCTTCACTTCGGGTCGG	27	
Duet-Hol_ΔSP_	TATGAATTCGCCCTGCCGCTGTGGA	25	

## Data Availability

The data used to support the findings of this study are available from the corresponding author upon request.

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
