# Peer review of "Characterization of the LysP2110-HolP2110 Lysis System in Ralstonia solanacearum Phage P2110"

_ijms, 2023, doi:10.3390/ijms241210375_

Round 1

Reviewer 1 Report

In the manuscript "Characterization of the LysP2110-HolP2110 Lysis System in Ralstonia solanacearum Phage P2110 C. Kaihong et. al. reported on the study of phage lysozymes infecting Ralstonia solanacearum.

There are significant errors and issues in the work that require serious correction before publication in the journal.

• Whole text: carefully check italics, eg lines 291, 293, 303 and bring all taxon names in the same form, add spaces before links in the text according to the requirements of the journal, improve the quality of pictures (eg picture 10b should be reduced in height).

• Figure 8: Why does the figure only show the standard error on the positive side of the mean? Fix it.

• Figure 13: Why did the bacteria initially have different titers? This complicates the interpretation of the results. Alternatively, give the % reduction in titer compared to control. Perform statistical processing.

• Figire 15: Why are the statistical groupings shown in only three dots (a b c)? Please justify this or do full statistical processing.

• Section Discussion: No plans to continue research. The practical meaning of adding sodium azide is not clear, and in general, how the authors are going to compose the formulation and use it to protect against the pathogen. Besides? the authors did not provide information on the safety of using lysozymes on the plant and the absence of their phytotoxicity. If these points were fulfilled, the work would be considered complete and complete.

• Table 3: Provide Genbank acsession numbers or papers that describe this strain of viruses and bacteria.

• Table 3: (Laboratory collection) - It is not clear which laboratory is being referred to.

Line 77

What do you mean by “modification proteins“? “Nucleic acid manipulations“?

Line 78

Quality of picture is not acceptable. Please improve the resolution. Please change “packing protein” to “packaging proteins”, “structure proteins” to “structural proteins” or “morphogenesis proteins”. Tyrosine recombinase is definitely not a packaging protein.

Lines 81-83

All tailed phages belong to the class Caudoviricetes. Please conduct a better taxonomic analysis to elucidate closest lower-ranked taxa including genera. Besides, this analysis can not give you definite conclusions about the host. “Erwinia amylovora” should be written in Italics.

Lines 86-87 including Table 1 and Figure 2

The organisms’ names should be written in Italics.

Lines 88-89

Please give a more detailed and correct caption to the picture and check typos.

Line 115, Figure 3

What does “Promote” mean? Do you mean the promoter region? How was it identified? Also, to make better conclusions, I would recommend to predict the holin structure using AlphaFold2.

Lines 129-131

The organisms’ names should be written in Italics.

Lines 134-139

Please make the assessment of the structural prediction quality. Again, I would recommend to use AlphaFold2 for structural predictions. At the moment your conclusions do not look reliable.

• Line 179: It is unclear which strain or strains were used in the test. Please add information to the materials and methods section.

• Line 526: Change 200rpm/min to 200 rpm because rpm is Rounds Per Minutes,

• Section 4.9: Specify the criterion by which the statistical difference was calculated? Also indicate the version of the program.

• Section 5: In this section it is worth mentioning the prospects for research and future plans. In addition, how the authors are going to apply them in defense against the pathogen.

lines 444-445

How did you choose the substitution model? Please give the setting of the phylogenetic tree inference tool used.

Author Response

Dear reviewer,

Thank you very much for your comments and professional advice. These opinions help to improve academic rigor of our article . Based on your suggestion and request, We have made corrected modifications on the revised manuscript. We hope that our work can be improved again. Furthermore, we would like to show the details in the PDF.

Thank you very much for your attention and time. Look forward to hearing from you.

Your Sincerely,

Kai Hong Chen

June 14,2023

Research “Characterization of the LysP2110-HolP2110 Lysis System in Ralstonia solanacearum Phage P2110 “(IJMS-2429315)

Reviewer 2 Report

The manuscript describes the dissection of the antibacterial effect of two proteins from a phage that is pathogenic to the bacterial plant pathogen Ralstonia solanacearum.

The experimental design is mostly appropriate (see some comments below) and the results lead towards definite conclusion regarding the effect of these two proteins and their synergistic effect.

Specific issues I found with the study are as follows:

1) The information on the phage P2110 (and the others cited in materials and methods) are a bit confusing. In Table 1 they are reported as "Ralstonia phage", but in line 83 it is stated that P2110 infects Erwinia amylovora. I believe that this point should be explained more thoroughly as it is quite confusing. If natural hosts of these phages are known and are different, these indications might be given to better contextualize the phylogenetic tree in Figure 2.

2) Figure 8: Why are there no indications for the control in panel B? I believe it would be interesting to compare also this value between the control (presumably close to 0?) and 28b-Lys mutant.

3)Figure 13. I believe that in this graph some graphical adjustments could be made to give a visual reference of which are R. solanacearum strains - the theoretical targets of this study - and which are the "control" bacteria. Also, while is the column for E. coli called Coil instead? The text (line 266) cites Pseudmonas syringae, but the graph does not have this bacterium.

4) Paragraph 2.7: from the presented results, there is no evidence that in the mutant that produces Hol without the signal peptide the protein is produced at all. The authors should provide evidence (another Western blot) that shows the production of a protein of expected size.

5) Figure 16. From the caption of the figure and the text it is unclear what is the difference between "control" and "0mM". What does "Empty cells" mean (line 325)? And how much Sodium azide was given to them? It is quite confusing and this should be explained better.

6) In the discussion of the results, it is simply stated that endolysin is effective against R. solanacearum. Still, based on the results shown in the manuscript the Po strain (R. solanacearum isolated from eucalyptus) shows results in line with Bacillus subtilis. This must be discussed.

7) Paragraph 4.1; this paragraph is simply unacceptable in the level of detail given. Table 3 reports only a code and that the strains are "Bacteria" without indication of a species or other taxonomic group. Also, the Description is confusing... what does it mean that the Bacteria are Wild Mushrooms? I imagined that could be the source of isolation, but that would in contrast with what is stated previously, giving different plant hosts for Ralstonia isolates. Also, strains ED3 and DH5a are not from this study, but rather from collections.

8) In most of Materials and Methods there sentences that speak about "The bacteria". I believe that in most cases the authors mean E. coli, and they should specify that precisely, also which strain they mean each time. Likewise, at the start of paragraph 4.7 the authors just talk about "Recombinant cells" but do no indicate which mutant they mean. Each mutant as well should be given a precise code used in every instance to identify it precisely throughout the text.

9) Based on what is presented in materials and methods, it seems that only Lys was subjected to Western Blotting, but in the results the blot for Hol is presented as well.

The language and form of the manuscript need special care as many sentences sport syntax mistakes (e.g. lack of concordance between subject and verb) and throughout the text many names are not italicized (especially bacterial names, such as E. coli o Erwinia amylovora). Also there are some typos, such as Nandrop instead of Nanodrop. Please carefully review all the manuscript, with help from a native English speaker if possible.

Author Response

(The authors gave the same response as above.)

Round 2

Reviewer 2 Report

I thank you for the corrections made to the article. I am satisfied with the changes and have no further comments.